# The Impact of Neurotransmitters on the Neurobiology of Neurodegenerative Diseases

**DOI:** 10.3390/ijms242015340

**Published:** 2023-10-19

**Authors:** Sarah E. Davis, Abagail B. Cirincione, Ana Catya Jimenez-Torres, Jun Zhu

**Affiliations:** Department of Drug Discovery and Biomedical Sciences, College of Pharmacy, University of South Carolina, 715 Sumter Street, Columbia, SC 29208, USA; st20@email.sc.edu (S.E.D.); abagailc@email.sc.edu (A.B.C.); aj83@mailbox.sc.edu (A.C.J.-T.)

**Keywords:** HIV-1 Associated Neurocognitive Disorders, Dopamine, neurotransmitters, neurodegenerative diseases

## Abstract

Neurodegenerative diseases affect millions of people worldwide. Neurodegenerative diseases result from progressive damage to nerve cells in the brain or peripheral nervous system connections that are essential for cognition, coordination, strength, sensation, and mobility. Dysfunction of these brain and nerve functions is associated with Alzheimer’s disease, Parkinson’s disease, Huntington’s disease, Amyotrophic lateral sclerosis, and motor neuron disease. In addition to these, 50% of people living with HIV develop a spectrum of cognitive, motor, and/or mood problems collectively referred to as HIV-Associated Neurocognitive Disorders (HAND) despite the widespread use of a combination of antiretroviral therapies. Neuroinflammation and neurotransmitter systems have a pathological correlation and play a critical role in developing neurodegenerative diseases. Each of these diseases has a unique pattern of dysregulation of the neurotransmitter system, which has been attributed to different forms of cell-specific neuronal loss. In this review, we will focus on a discussion of the regulation of dopaminergic and cholinergic systems, which are more commonly disturbed in neurodegenerative disorders. Additionally, we will provide evidence for the hypothesis that disturbances in neurotransmission contribute to the neuronal loss observed in neurodegenerative disorders. Further, we will highlight the critical role of dopamine as a mediator of neuronal injury and loss in the context of NeuroHIV. This review will highlight the need to further investigate neurotransmission systems for their role in the etiology of neurodegenerative disorders.

## 1. Introduction

The prevalence of neurodegenerative disorders (NDs) is increasing in the United States. It is projected that 13.8 million people will be affected by Alzheimer’s disease (AD) by 2050 [1]. Additionally, the incidence of Parkinson’s disease (PD) has increased from 1990 to 2019 [2]. FDA-approved medications for alleviating the symptoms of AD and PD involve the rebalance of neurotransmitter (NT) systems. For instance, in AD patients, acetylcholine esterase inhibitors are used to increase acetyl choline (ACh) available for signaling [3]. Moreover, levodopa (L-DOPA), along with monoamine oxidase (MAO) inhibitors and dopamine (DA) agonists, have been used clinically as major medications for the treatment of symptoms associated with PD [4]. The current generation of medication in development for NDs is being designed to target underlying disease pathogenesis and pathology rather than alleviating disease-induced symptoms. For the treatment of AD, Aducanumab was approved for medical use in the United States by FDA in 2021. Aducanumab is a medication used in the management and treatment of AD, which is an amyloid-β (Aβ) -directed monoclonal antibody [5]. However, the use of medications targeting Aβ is controversial, as this class of drugs has not been highly efficacious at rescuing cognitive decline in AD patients. Regardless, the development of drugs that target the underlying etiology of NDs, which function to prevent neuronal injury, is of great clinical interest.

While the etiology of different NDs varies, a unifying feature of NDs is the activation of downstream pathways that activate cellular death, including but not limited to apoptosis and pyroptosis. Activation of these pathways is brought about by neurotoxic molecules which promote inflammation and/or mitochondrial dysfunction. Importantly, dysregulation of NTs induces such neurotoxic molecules. For example, methamphetamine (meth) dysregulates DA homeostasis via interaction with the DA transporter (DAT), leading to autooxidation of DA and the production of neurotoxic reactive oxygen species (ROS). The meth-induced production of ROS from dysregulation of DA metabolism has been directly linked as the cause of meth-induced neurotoxicity [6].

We propose there are two paths underlying NT-mediated neurotoxicity. First, the direct path involves neurotoxic NT metabolites, which can directly influence cell stress response pathways. As described above, neurotoxicity and cell death induced by meth represents the direct pathway for NT-mediated neurotoxicity. This phenomenon has been observed in the context of PD, where inappropriately formed DA metabolites in the brain were found to be directly responsible for mediating alpha-synuclein (αSyn) induced cell—death [7]. Second, the indirect path involves the overactivation of NT receptors by dysregulated NT systems. Specifically, the inappropriate activation of G-protein coupled receptors (GPCRs) can increase intracellular Ca^2+^ levels. Increased Ca^2+^ can then perturb mitochondrial function and activate mitochondrial-induced apoptosis [8]. In this review, we will discuss the dopaminergic and cholinergic neurotransmission, two NT systems commonly dysregulated in NDs. Further, we will discuss evidence for both pyroptosis and mitochondrial-mediated apoptosis in NDs. We will also review the possible mechanisms by which dysregulated NT systems observed in NDs can activate downstream pathways to bring about neuronal death. Lastly, we will highlight the role of DA as a critical mediator of both synaptodendritic injuries as well as neuronal loss observed in HIV-positive persons. With this review, we aim to shed light on how the dysregulation of NT systems impacts the neurobiology of NDs.

## 2. Dopaminergic Neurotransmission

### 2.1. Regulation of Dopamine Synthesis

DA is a neurotransmitter that is involved in several pathways in the brain, including the nigrostriatal, mesolimbic, and mesocortical projections. The production of DA, its vesicular localization and release, and its extracellular persistence are regulated by the activity of tyrosine hydroxylase (TH), the vesicular monoamine transporter (VMAT-2), and the DAT [9]. DA is produced from the aromatic amino acid tyrosine, after which it is converted by the rate-limiting enzyme TH into L-3,4-dihydroxyphenylalanine (L-DOPA). L-dopa is then converted into DA via the enzyme aromatic amino acid decarboxylase (AADC) [10]. Once in dopaminergic terminals, DA molecules are sequestered and packaged into secretory vesicles via transport proteins [11]. The signal termination and degradation mechanisms for DA transmission include enzymatic processing by monoamine oxidase (MAO-B), which is located inside of the presynaptic terminal, and catechol-o-methyltransferase (COMT), which is located inside the synaptic cleft [9,12]. When DA is released into the synapse, it can bind to both presynaptic and postsynaptic DA receptors, after which it is transported back into the presynaptic terminal by the DAT. Once in the cytosol, DA will either be repackaged into synaptic vesicles or become degraded [13].

### 2.2. Dopamine Transporter Regulation

DA receptors can be divided into either D1- or D2-like receptor families. The D1 type family includes G protein-coupled receptors D1 and D5, which are excitatory and expressed post-synaptically. D1-like receptors are coupled to the stimulatory G-protein Gs and can induce excitatory transmission by upregulating adenylate cyclase activity and subsequent production of cyclic AMP (cAMP). Conversely, the D2-like family includes D2, D3, and D4 receptors, which are expressed in both presynaptic and postsynaptic terminals. D2-like receptors are coupled to the G protein Gi and induce inhibitory transmission by reducing cAMP production via inhibition of adenylate cyclase activity [14]. DA signaling is primarily terminated by the reuptake of DA by DAT into the cytosol and then by VMAT-2, which transports DA from the cytosol into synaptic vesicles [9]. DAT is a member of the Na^+^/Cl^−^ coupled neurotransmitter solute carrier 6 (SLC6) family and is highly conserved among humans and other mammals [9,12]. The DAT consists of 12 transmembrane helices and large N and C terminal tails located intracellularly, which contain sites for post-translational modifications [15,16]. The kinetics of DA transport through DAT follow the Michaelis-Menten model [17]. DAT transports DA into the cytosol by a channel-like mechanism involving the transition between inward-facing and outward-facing confirmations [12,15,18]. This alternation of conformations is accomplished by the symport of two sodium ions and one chloride ion into the cytosol along their concentration gradients, which are established by the Na^+^/K^+^-ATPase [19]. First, DAT occupies the “outward facing” conformation wherein substrates bind to their uptake sites on the extracellular side of the transporter. Once the substrate binds the DAT, a conformational change occurs wherein DAT occupies the “inward facing” conformation, and the translocates into the neuron.

Dopaminergic neurotransmission is altered by DAT-binding pharmacological agents which bind near the DA uptake site, including cocaine and meth [12,17]. Specifically, cocaine competitively inhibits the DA uptake site on DAT and induces DAT conformational transition at the outward-facing stage, leading to increased extracellular DA levels. Conversely, meth-like drugs act as a DAT substrate and stabilize the inward-facing conformation of DAT, thus causing the efflux of DA through DAT from the intracellular to the extracellular space. While these drugs interact with DAT near the DA uptake site, modulation of DAT function at allosteric binding sites also alters DA transport dynamics. Currently, the development of allosteric modulators for the DAT is being investigated for their therapeutic potential for substance use disorder and comorbid HIV-1 infection [20,21].

Post-translational modification of DAT by phosphorylation is important for modifying DAT function [9,22]. Specifically, the four threonine (Thr43, 46, 48, 62) and eight serine (Ser 2, 4, 7,13, 14, 44, 45, 64) residues on the N terminus of DAT can undergo phosphorylation by protein kinase C (PKC), protein kinase A (PKA), protein kinase G, and Ca^2+^ calmodulin protein kinase II [9,23]. PKC-mediated phosphorylation of DAT regulates DAT endocytosis, transport, and efflux. Specifically, PKC can phosphorylate serine residues on the intracellular N-terminus tail [15,22,24]. PKC-mediated phosphorylation of DAT changes DAT function by stimulating clathrin and dynamin-mediated endocytosis to remove DAT from the surface of the plasma membrane [25,26]. Another kinase that may modify DAT function is the extracellular signal-regulated Kinase (ERK). ERK phosphorylates the proline residue next to Thr53, thus altering DAT structure and increasing DAT’s DA transport capacity [15].

Other post-translational biochemical modifications to DAT can also regulate the protein’s function, including glycosylation, palmitoylation, and ubiquitination. Glycosylation of asparagine residues located on the extracellular loop of DAT helps to stabilize DAT localization to the plasma membrane and increases DA uptake via DAT. Further, glycosylation of DAT is increased with age and has been shown to play an important role in the susceptibility of substance use disorders [9]. Palmitoylation of C-terminus residues to add saturated fatty acids to DAT functions to reduce PKC-mediated endocytosis of DAT. Specifically, palmitoyl acyltransferases known as DHHC enzymes (2, 3, 8, 15 and 17) covalently attach a palmitoyl group via a thioester bond to the Cys580 reside of DAT on the C-terminus. This, in turn, stabilizes DAT expression and uptake kinetics [27]. Lastly, ubiquitination of the lysine residues on the N terminus by ubiquitin E3 ligases Nedd4-2 and Parkin regulates the recycling and degradation of DAT [22].

In addition to biochemical modifications, DAT function is also regulated through protein–protein interactions, such a D2 receptors (D2R) and sigma receptors. D2R expressed in presynaptic terminals can interact with the N-terminus of DAT via its intracellular loop. D2R interaction with DAT has been shown to enhance DA uptake and facilitate recruitment of DAT to the plasma membrane [28], and further, has been attributed to DA-induced neurotoxicity [29]. Sigma receptors expressed in dopaminergic regions have been shown to affect DA synthesis, release, and reuptake in the substantia nigra (SNr) and the ventral tegmental area (VTA) [30,31]. There are two established sigma receptor types, σ_1_R and σ_2_R. The σ_1_R functions to regulate Ca^2+^ homeostasis; however, σ_1_R, which is generally localized near the plasma membrane, has been shown to regulate several different transmembrane protein activities through direct protein–protein interaction. Regarding dopaminergic proteins, σ_1_R has been shown to directly modulate D1 and D2 receptors, as well as DAT. The impact of σ_1_R interaction with DAT on DA transport remains unclear, but evidence does suggest that σ_1_R activation may increase DA uptake through DAT [31]. Considering the importance of σ_1_R in mediating the effects of DAT inhibitors such as meth on DA release, characterizing the σ_1_R/DAT interaction is of significant interest [32].

### 2.3. Vesicular Monoamine Transporter Regulation

Vesicular monoamine transporters, including VMAT-1 and VMAT-2, belong to the SLC18 family of vesicular amine transporters. VMAT-1 is primarily expressed in neuroendocrine cells, whereas VMAT-2 is expressed in the central peripheral and enteric nervous systems and is localized in presynaptic terminals [9,33]. VMAT-2 plays a critical role in sorting, storing, and releasing NTs in order to protect neurons by counteracting intracellular toxicity [34,35]. Generally, VMAT-2 inhibitors are classified as being either “reserpine-” (irreversible) or “tetrabenazine” (TBZ)-like [36,37]. VMAT-2 is critical for mediating the psychoactive effects of amphetamines. Amphetamines both inhibit DA uptake through VMAT-2 at reserpine sites and release DA from vesicles into the cytosol, which is then reverse transported through DAT [38]. The loading of NTs into vesicles by VMAT-2 occurs against their concentration gradient. Therefore, VMAT-2 functions as an H^+^-antiporter, releasing two protons from the vesicle for every monoamine translocated into the vesicle. This process is accomplished using proton gradients, which are established using H^+^- ATPases located on the lumen of vesicles [35]. Amphetamine-induced release of DA from vesicles is a result of disrupting this proton gradient.

VMAT-2 localization and function are regulated by post-translational modifications, including glycosylation and phosphorylation. VMAT-2 is synthesized in the ER and then glycosylated in the Golgi apparatus. Colocalization of the N-linked glycosylation loop (located between TM1 and TM2) and the C-termini tail of VMAT-2 is necessary for the localization of VMAT-2 to large dense core vesicles [39]. Further, phosphorylation of serine residues near the C-terminus of VMAT-2 by serine/threonine-directed kinase CKII is also thought to contribute to VMAT-2 localization to large dense core vesicles, as evidenced in PC12 cells [40]. Lastly, phosphorylation is also suspected to regulate VMAT-2 function. Phosphorylation of N-terminal residues Ser15 and Ser18 by PKC was found to be necessary for meth-stimulated efflux of 5-HT [41]. Thus, the N-terminal and C-terminal domains of VMAT-2 located in the cytoplasm are important for regulating VMAT-2 localization as well as function.

## 3. Cholinergic Neurotransmission

### 3.1. Regulation of Acetylcholine Transmission

Cholinergic neurotransmission is essential for cognitive function, synaptic plasticity, and memory [42,43]. Acetylcholine (ACh) is released by cholinergic neurons, which are primarily located in the subcortical regions with projections to the cortical areas [44,45]. Choline acetyltransferase (ChaT) is responsible for ACh synthesis and is required for cholinergic neurotransmission in the central and peripheral nervous system [46]. ChaT is produced in the soma and then localized mainly in the axon terminal, where it catalyzes the synthesis of ACh from choline and acetyl-CoA [44,47]. Like DA release, the exocytotic release of ACh is dependent on the SNARE protein complex [48]. After exocytotic release, ACh interacts with cholinergic receptors to relay information in the CNS and PNS and mediate synaptic transmission at neuromuscular junctions [49]. Acetylcholinesterase (AChE), primarily found at postsynaptic neuromuscular junctions, metabolizes ACh in the extracellular space into acetate and choline, thus terminating the ACh signal [50]. Inhibition of AChE causes an accumulation of ACh in the synaptic cleft and continuous activation of the cholinergic receptors. Therefore, AChE inhibitors are used in various pharmacological treatments, including the treatment of AD, which is characterized by deficits in ACh levels [44]. However, inappropriate inhibition of AChE has negative consequences, as increased ACh can lead to overstimulation of cholinergic receptors and have harmful effects [51]. Finally, released choline in the synaptic cleft is transported back into synaptic terminals by the choline transporter (CHT). The CHT is a sodium-coupled transporter containing 13 transmembrane domains and belongs to the SLC5A7 family of glucose transporters. Regulation of choline uptake is primarily accomplished through the trafficking of CHT to the plasma membrane. CHT recycling occurs via clathrin-mediated endocytosis, wherein endosomes either reenter the synaptic vesicle cycle or mature into lysosomes, upon which CHT is then degraded [52].

### 3.2. Acetylcholine Signal Transduction

There are two types of cholinergic receptors: nicotinic and muscarinic receptors [51]. Nicotinic acetylcholine receptors (nAChRs) are found in the central and peripheral nervous system and neuromuscular junction [51,53]. nAChRs are formed by the assembly of five transmembrane subunits. Here, we mainly discuss neuronal nAChRs, which are assembled either as homo-pentamers of α7, α8, and α9 or hetero-pentamers of α2–α6 in combination with β2–β4 or α9 with α10 subunits [54,55].

Activation of nAChRs initiates the opening of an ion pore channel, allowing sodium and calcium influx and potassium efflux across the cell membrane, which in turn regulates neuronal membrane potential and excitability [42]. The nAChRs-mediated ion channel activation initiates a tertiary conformational transition of nAChRs, which includes distinct resting, open, and desensitized states. These conformational state transitions are dynamic and depend on nAChRs subtype and agonist concentration [56]. The most predominant subtypes of nAChRs expressed in the human brain are the heteromeric α4β2 and the homomeric α7 [57,58]. The α4β2 subtype is important for mediating nicotinic effects on DA in the mesolimbic system, whereas the α7 subtype is involved in glutamatergic and dopaminergic release in the CNS. Both α4β2 and α7 subtypes contribute to the pathogenesis of a range of neurological disorders, including AD, schizophrenia, PD, and depression [59,60,61].

The muscarinic receptor family includes five different metabotropic G-coupled protein receptors involved in the parasympathetic nervous system in the brain, which can activate a multitude of signaling pathways important for the modulation of neuronal excitability, synaptic plasticity, and feedback regulation of ACh release [44,51]. Receptors M_1_, M_3_, and M_5_ are coupled with a Gq subunit, leading to IP_3_-mediated Ca^2+^ release through phospholipase C activation, while M_2_ and M_4_ receptors are coupled with a Gi subunit and function to inhibit adenylate cyclase and prevent the formation of cAMP [14]. For example, M_2_ can inhibit beta-adrenergic stimulated relaxation, and further, M_3_ plays a role in the control of the contraction of airway smooth muscle [62]. In the CNS, muscarinic receptors are located both pre- and post-synaptically on neurons. Expression patterns of muscarinic receptor subtypes vary across the CNS. Due to the role of ACh in cognition and substance use disorders, drugs that modulate the function of muscarinic receptors are of great interest and have been reviewed extensively [63].

### 3.3. Vesicular Acetylcholine Transporter Regulation

At the presynaptic terminal, synthesized ACh is packaged into synaptic vesicles for storage by the vesicular acetylcholine transporter (VAChT). VAChT is the third member of the SLC18 gene group, SLC18A3. The structure of VAChT includes 12 transmembrane domains, which are bundled into two groups (TM1–6 and TM7–12), with the N- and C-termini extending into the cytoplasm [64]. VAChT functions to transport ACh against its concentration gradient by coupling ACh transport to the efflux of two protons [65]. These protons are transported with their concentration gradient, which is established by an H^+^-ATpase. ACh release is triggered by Ca^2+^ influx through specific presynaptic Ca^2+^ channels. VAChT trafficking is regulated by a di-leucine motif on its C-terminus, which helps to localize VAChT to synaptic vesicles. One well-established drug that targets VAChT function is vesicamol, a cell-permeable non-competitive inhibitor that interacts with the C-terminus of VAChT to prevent repackaging of ACh into vesicles. In the context of NDs, decreased VAChT mRNA levels have been observed in patients with AD compared to healthy controls [66]. Decreased VAChT availability reduces the amount of ACh available for release, thus negatively impacting memory. Henceforth, changes in VAChT expression may underlie observed deficits in memory associated with AD.

## 4. Neurotransmitter Hypothesis of Neurodegeneration

### 4.1. Evidence for Pyroptosis in Neurodegenerative Diseases

One major form of cell death evidenced in NDs is the inflammatory-mediated process called pyroptosis [67]. Pyroptosis is caused by the activation of inflammatory caspase-1/4/5 or 11) and is distinct from traditional apoptosis, as pyroptosis leads to the release of inflammatory cytokines [68]. Specifically, the cytosolic pattern recognition receptor (PRR) protein NLRP3-mediated inflammasome has been implicated as a mediator for NDs [69,70]. In the context of PD, the release of the inflammatory cytokine IL-1β is promoted by the misfolded αSyn protein, a key hallmark of PD. IL-1β primes NLRP3-inflammasomes for activation by αSyn, thus producing pyroptosis [71]. The canonical proteins thought to mediate AD, Aβ, and phosphorylated Tau have also been shown to initiate NLRP3- and NLRP1-mediated neuronal pyroptosis [72,73]. Recent evidence identified cleaved Gasdermin D (GSDMD), a marker for pyroptosis, in microglia and astrocytes near local neuronal loss sites and Aβ plaques in post-mortem brain tissue of AD patients [74]. Lastly, Huntington’s disease (HD) is characterized by a repeat expansion of CAG (>35) in the *IT15* gene, which encodes for a protein associated with axonal transport [75]. Several forms of neuronal cell death, including necroptosis, ferroptosis, and apoptosis, have been implicated in HD. Recent evidence also suggests the role of pyroptosis in HD, as evidenced by reduced disease progression in an HD mouse model treated with the NLRP3 inhibitor MCC950 [76,77]. While multiple forms of neuronal cell death are involved in the neurodegeneration of NDs, activation of NLRP3 inflammasomes is of great interest in the pathophysiology of neurodegeneration.

### 4.2. Dysregulation of Neurotransmission in Neurodegenerative Disorders

AD is a neurodegenerative disorder characterized by memory loss and behavioral and psychological symptoms. An imbalance of different NTs, including glutamate (Glu), γ-Aminobutyric acid (GABA), ACh, DA, and 5-HT, is well evidenced in AD. For example, cholinergic neuronal loss and decreased molecules responsible for ACh metabolism and neurotransmission, such as acetylcholine esterase, are observed in AD [78]. Furthermore, monoaminergic systems are reduced in patients with AD, which may be due to increased MAO activity observed in at-risk persons carrying the ε4 allele of apolipoprotein E [79]. Specifically, disturbances in the dopaminergic system have been evidenced in both AD and PD. Moreover, patients with AD have reduced D1R and D2R expression [80] and selective neuronal loss in the VTA [78]. Regarding the major neurotransmitters in the CNS, Glu, and GABA, Aβ peptides can impact glutamatergic (excitatory) neurotransmission through several routes, including inhibition of Glu reuptake, enhanced Glu release, and dysregulation of Glu/glutamine synthesis [81]. Circulating levels of GABA, which are responsible for the inhibition of synaptic firing, are reduced in the CSF of AD patients. A feedback loop between GABA_A_ receptor activation and an increase in neurofibrillary tangles (NFTs) as a result of increased tau phosphorylation has been postulated as a contributor to AD pathology [82].

In PD, neuronal loss is restricted to dopaminergic neurons, primarily in the SNr [83]. Specifically, neurotoxic dopaminergic metabolites (primarily aminochrome and 5,6-indolequinone) are hypothesized to mediate dopaminergic loss due to the adducts they form with αSyn [84]. Indeed, DA metabolites and catabolites are a major source of oxidative stress observed in PD and are elevated in clinical CSF samples from PD patients compared to healthy controls [85]. Together, these present a possible role for dysregulation of DA homeostasis in the pathology of PD. Further, the serotonergic system is implicated in the difficulty of treating PD. While it is controversial whether PD pathology directly impacts 5-HT systems, it has been reported that the serotonergic system plays a vital role in L-DOPA-induced dyskinesias (LID) by promoting the non-physiological release of DA [86]. Understanding how serotonergic pathways influence dopaminergic release and motor function is of critical interest for improving therapeutic interventions for PD.

### 4.3. Evidence for Neurotransmission as a Mediator of Neurodegeneration

#### 4.3.1. Calcium Function in the Neuron

Due to its electrochemical nature and ability to interact with complex binding sites, Ca^2+^ plays a variety of roles in the body, serving as a versatile carrier of information in many signaling networks [87]. Here, we will mainly focus on Ca^2+^ as it functions in neurons and in the context of NDs. In neurons, Ca^2+^ regulates the exocytotic release of NTs from vesicles. Voltage-gated channels (Cav) expressed on the plasma membrane of axon terminals are activated in response to membrane depolarization. Upon activation, these channels allow for the influx of Ca^2+^ ions into cytosolic space. Various Cav channels have been identified and are characterized based on the threshold of depolarization required for their activation. In synapses, there are two primary Cavs responsible for Ca^2+^ influx, Cav 2.1 and Cav 2.2, and are differentiated by their specific α1 subunit composition (Cav 2.1 is α1A, Cav 2.2 is α1B) [88]. Vesicles that are “primed” at the plasma membrane in the active zone undergo fusion in response to Ca^2+^ binding. Specifically, Ca^2+^ causes synaptotagmin-1 to be released from the SNARE (soluble N -ethylmaleimide sensitive factor attachment protein receptor) complex. The synaptotagmin-1 interacts with the plasma membrane and cooperates with the SNARE complex to trigger vesicle fusion [89]. The SNARE complex forms within milliseconds due to “nanodomains” of localized increases in Ca^2+^, which occur after Cav activation. These local increases in Ca^2+^ are coupled to the active zone such that NT release may occur on a sub-second timescale. After the tripartite complex is formed, fusion occurs, at which point NTs are released via exocytosis into the synaptic cleft, completing the transmission of the signal [90].

In both neuron and non-neuronal cells, Ca^2+^ acts as a secondary messenger to carry information and mediate changes at the level of transcription. Although the role of Ca^2+^ in many different modulatory functions in cells has been reviewed previously [91], we will focus on the role of Ca^2+^ within the context of the GPCR (Gαq) activation pathway. Upon binding to a ligand, a guanosine diphosphate (GDP) bound to the α subunit of the GPCR is replaced with GTP, upon which the Gαq-GTP subunit dissociates from the GPCR complex to interact with downstream effector molecules [92]. The Gαq is inactivated by intrinsic hydrolysis of GTP to GDP, upon which the inactive Gαq-GDP complex binds to the GPCR again to await reactivation. The canonical pathway in which the Gαq-GTP effects changes in intracellular activity is through activation of phospholipase C (PLC) enzymatic hydrolysis of phospholipid phosphatidylinositol bisphosphate (PIP_2_) to Inositol-1,4,5-triphosphate (IP_3_) and diacylglycerol (DAG) [93]. DAG activates protein kinase C (PKC), which in turn can activate protein kinase D (PKD) through phosphorylation of PKD residues Ser744 and Ser748. PKD activation is associated with many different cellular functions, including the promotion of antiapoptotic pathways [94]. On the other hand, IP_3_ functions to mobilize the release of intracellular stores of Ca^2+^ from the ER through the activation of IP_3_ receptors (IP_3_Rs). Intracellular stores of Ca^2+^ in the ER are maintained in the millimolar range, whereas cytosolic levels of Ca^2+^ generally do not exceed 100 nM. Two major types of channels are expressed on the ER, which are Ca^2+^ permeable: ryanodine receptors (RyR) and IP_3_ receptors (IP_3_R). When activated, these channels increase cytosolic Ca^2+^ concentrations. The function and structure of RyRs and IP_3_Rs have been reviewed elsewhere [95]. To return the cytosol back to its low-level Ca^2+^ state, Ca^2+^ ions are transported back into the ER for storage by sarcoendoplasmic reticulum calcium ATPase (SERCA) pumps. Additionally, negative feedback is provided by the elevated Ca^2+^, which can subsequently regulate this effect by inhibiting the opening of the RyR and IP_3_R channels. Further, Ca^2+^, along with DAG, can activate PKC-mediated phosphorylation of PLC. This, in turn, reduces PLC activity, thus providing an additional level of negative regulatory feedback for intracellular Ca^2+^ release from the ER.

#### 4.3.2. Mitochondrial Function in the Neuron

Mitochondria are double-walled organelles that hold the electron transport chain and play host to the final steps of anaerobic respiration. It is estimated that a single neuron may contain millions of mitochondria [96]. This is necessary, as the brain relies primarily on mitochondria for ATP production as its energy source. In neurons, mitochondria are important for tight regulation of intracellular Ca^2+^ levels. Mitochondria interact with the ER at specific sites called mitochondria-associated membranes (MAMs). It is at these sites where Ca^2+^ (released from the ER via IP_3_Rs) can be transported into the mitochondria via the mitochondrial calcium uniporter (MCU) [97]. The coupling of Ca^2+^ release from the ER to uptake into the mitochondria at MAMs functions to provide Ca^2+^ for calcium-dependent enzymes functioning in the Krebs Cycle during anaerobic respiration. While Ca^2+^ transport to the mitochondria is necessary for meeting bioenergetic demands, excessive Ca^2+^ in the mitochondria is not conducive to cell survival [98]. When mitochondrial Ca^2+^ levels are excessive, mitochondrial permeability transition pores open (mPTPs) [99]. The formation of mPTPs causes depolarization of the mitochondrial membrane and swelling of the mitochondria, thus releasing pro-apoptotic proteins. Specifically, depolarization of the outer mitochondria membrane releases apoptotic peptidase activating factor 1 (APAF1), which activates caspase 9, to then activate downstream “executioner caspases” 3 and 7. Additionally, Ca^2+^ regulates mitochondrial depolarization by regulating the activation of the pro-apoptotic family of B cell lymphoma (Bcl-2) proteins, which contains both pro- (Bax) and anti- (Bcl-2) apoptotic proteins (Figure 1) [100].

#### 4.3.3. Neurotransmitter Mediated Dysregulation of Ca^2+^

Various neurotransmission systems are disrupted in NDs. Independent from this, mitochondrial function is highly dependent on homeostasis Ca^2+^, and disruption of mitochondrial function (with or without Ca^2+^ disturbances) can mediate mitochondrial-induced apoptotic pathways. Therefore, understanding how NTs disrupt intracellular Ca^2+^ to then (indirectly) initiate mitochondrial-mediated apoptosis is of great interest.

Extracellular levels of NTs and intracellular levels of Ca^2+^ regulate one another. As described previously, local influxes of Ca^2+^ via Cav channels are required for the exocytotic release of NTs into the synaptic cleft for neurotransmission. Conversely, NTs can regulate intracellular Ca^2+^ levels through activation of GPCRs, which are expressed pre- or post-synaptically, as well as near the soma. GPCRs regulate NT release through modulation of Ca^2+^, primarily through inhibition of voltage-gated calcium channels by Gβγ subunits [101]. However, more importantly, NTs can activate GPCRs coupled to Gαq/s can increase intracellular Ca^2+^ levels [14] by generating IP_3_ (Gαq path) to release intracellular stores of Ca^2+^ [102]. There are many GPCRs coupled to Gαq that are involved in the transmission of signals across synapses. Some examples include metabotropic Glu receptors (mGluRs) 1 and 5, heteromers D1R-D2R and D2R-D5R, mAChR-1, 5-HT-R 2C, α-1 adrenergic receptors, and heteromeric GABA_B(1,2)_ receptors [103,104,105,106,107,108]. Considering each NT system contains at least one Gαq coupled receptor in its toolbox of receptors used to transduce its signal, there are many possible pathways by which NTs can dysregulate intracellular Ca^2+^ levels through overactivation of GPCRs. Considering the role of GPCRs in regulating internal Ca^2+^ levels and the importance of Ca^2+^ in promoting mitochondrial-induced apoptosis, overactivation of GPCRs by disturbances in NT levels is a major concern for the pathophysiology of NDs [109].

In AD, increased intracellular Ca^2+^ is regulated by several factors, including GPCRs such as mGluRs and mAChRs [110,111]. Indeed, increased ROS and reduced mitochondrial function and size are observed in AD [112]. In the context of PD, neurotoxic elevated cytosolic DA is regulated in the SNr by Ca^2+^ steady-state levels [7]. In the SNr, mitochondria must work to meet a high bioenergetic demand due to the SNr’s extensive axonal arbor. This is accomplished in part due to the coupling of Cav1.3 channel-mediated Ca^2+^ release to mitochondrial oxidative phosphorylation. However, while this feature helps SNr neurons to meet energy demands, it puts these neurons at higher risk for mitochondrial stress and ROS [113]. Independent of NTs, αSyn aggregation, the hallmark of PD pathology, increases with Ca^2+^ levels. IP_3_ kinase B (ITPKB), which inactivates IP_3_, a secondary messenger that increases the release of Ca^2+^ from ER stores, was found to negatively regulate αSyn aggregation. Considering how NTs and Ca^2+^ levels regulate one another, disruption of NTs to increase cytosolic Ca^2+^ may contribute to αSyn aggregation. Understanding how disturbances in NTs may translate to changes in GPCR activation is needed to understand how NTs may contribute to the neurobiology of AD and PD.

## 5. Neurodegeneration and HIV-1 Associated Neurocognitive Disorders

### 5.1. NeuroHIV Background

HIV-1-associated neurocognitive disorders collectively referred to as “HAND” include asymptomatic neurocognitive impairment (ANI), HIV-associated mild neurocognitive disorder (MND), and HIV-associated dementia (HAD) [114]. The most severe form of HAND, HAD, includes Parkinsonian-like symptoms such as bradykinesia, rigidity, and hypophonia [115]. In addition to the other NDs discussed in this review, NLRP3 inflammasomes have been evidenced in the neurobiology of NeuroHIV. The HIV-1 protein transactivator of transcription (Tat) induces expression of microglial NLRP3 and IL1β, which results in synaptodendritic injury to neurons [116]. Another HIV-1 protein tied to HAND progression, glycoprotein 120 (gp120), also has been shown to promote neuroinflammation and death through microglial NLRP3 and IL1β mediated pyroptosis [117]. NLRP3-mediated inflammasome-induced pyroptosis is a major pathway toward neuronal loss, which should be characterized further in the context of NeuroHIV.

### 5.2. Disturbances in Neurotransmitter Systems in HAND

Dopaminergic neurotransmission has been implicated as a mediator of HAND pathology. Brain volume measured in patients with HAD post-mortem showed a reduced volume of basal ganglia, which correlated with the severity of HAD [118]. Further, clinical reports of DA in the CSF of HIV-1 infected persons show increased DA levels in early disease [119,120] and decreased DA in late disease stages [121,122,123], even when HIV-1 viremia is suppressed under combined antiretroviral therapy (cART) treatment [124,125]. Combined computational modeling and mutagenesis approaches identified the HIV-1 protein Tat as a negative allosteric modulator of the DA transporter (DAT) [126,127], as well as the NE transporter (NET) [128,129,130]. Acute expression of physiologically relevant levels of Tat [131,132,133,134] increases phasic-like vesicular release of DA, an effect that can be reversed by a novel allosteric modulator of DAT in inducible transgenic mice [20,135]. In addition to Tat, gp120 has also been shown to disturb monoamine systems through inhibition of DA uptake through DAT in various cell models [136,137,138]. Unsurprisingly, clinical reports using PET scan imaging found decreased expression of DAT in patients with HAD, independent of comorbid substance use status [139]. Indeed, disturbances in DA function are an important feature in NeuroHIV, which persists even in the post-cART era.

### 5.3. Evidence for Mitochondria and Ca^2+^ Disturbances in HAND

It has long been evidenced that Tat and gp120 are neurotoxic proteins, inducing neuronal apoptosis [140,141,142] preferentially in dopaminergic neurons [136,143,144]. Tat and gp120 have long been shown to disturb Ca^2+^ homeostasis by initiating IP_3_-mediated Ca^2+^ release from ER stores [145,146]. Further, Tat has been shown to initiate mitochondrial-induced apoptosis through Ca^2+^-mediated mitochondrial generation of ROS and caspase activation in hippocampal neurons [147]. Therefore, Tat-induced disturbances in Ca^2+^ homeostasis, which induce mitochondrial-mediated neurotoxicity, in addition to NLRP3 inflammasome-mediated pyroptosis, may also be a major contributor to the neurobiology of HAND [148].

While downstream effects of Tat on Ca^2+^ disturbances are well evidenced at this point, mechanisms by which these two key neurotoxic HIV-1 proteins initiate these effects are still coming to light. Tat-induced increases in intracellular Ca^2+^ levels have been attributed to the overactivation of L-type Ca^2+^ channels [149], as well as the potentiation of N-methyl-D-aspartate receptor (NMDAR) mediated Ca^2+^ influx [150]. In cortical neurons, Tat induces Ca^2+^ release from ER stores through a ryanodine receptor (RyR) dependent mechanism [151]. Although these reports do well to show that Tat does indeed disrupt Ca^2+^ through multiple Ca^2+^ regulatory mechanisms, it is not known whether Tat directly interacts with these channels to alter their function. Tat-induced disturbances of Ca^2+^ have been considered as the underlying mechanism for Tat-induced alterations in NT systems [152], including not only DA systems but GABA [153] and ACh as well [154]. It would not be surprising if Ca^2+^ dysregulation contributed to the Tat-induced disruption of NT systems, considering the critical role of Ca^2+^ in exocytotic NT release. However, direct interaction between Tat and molecular players has, to our knowledge, only been confirmed with monoamine transporters DAT and NET [130,155]. Therefore, the inverse relationship, wherein Tat-induced disturbances in Ca^2+^ homeostasis are downstream effects of Tat-induced disturbances in NT systems, we argue, is a possibility. NTs can dysregulate Ca^2+^ homeostasis by signaling through GPCRs and downstream signaling cascades triggered by activation of specific GPCR subtypes. Tat-induced effects on ACh release in cortical synaptosomes were shown to involve the mGluR1. These receptors, upon activation, provided regulatory feedback to increase Ca^2+^ release from the ER, increase Ca^2+^ influx from NMDARs, and finally increase noradrenaline release [156]. Additionally, Tat alters the expression and activity of DA receptors, another family of GPCRs. Expression of various DA receptors (D1, D2, D4, and D5) is reduced in the NAc of iTat-tg mice [157]. Further, Tat increases the excitability of D1 medium spiny neurons in the NAc by triggering IP_3_-mediated release of Ca^2+^ to depolarize the neuronal membrane [158]. Independent of Tat, extracellular DA can increase IP_3_-mediated Ca^2+^ release in macrophages via D5 receptors coupled to Gq proteins [159]. DA has also been shown to increase the release of inflammatory cytokines from human macrophages [160]. Considering the ability of Tat to increase extracellular DA release in the striatum [135], the aforementioned studies present two different mechanisms by which Tat-induced perturbances in extracellular DA may contribute to dopaminergic-specific neuronal damage observed in HAND. Specifically, Tat-induced disruption of DA neurotransmission may initiate neuronal loss not only due to mitochondrial-associated apoptosis via GPCR-mediated release of internal Ca^2+^ stores but also due to DA-induced neuroinflammation and downstream activation of pyroptosis in response to cytokine release from macrophages. Thus, preventing the effects of proteins such as Tat from disturbing NT systems is of great interest in mitigating the neuronal loss and damage underlying the neurobiology of HAND (Figure 2).

### 5.4. Mechanisms for Synaptic Injury

During the era of cART, the incidence of HAND has remained steady; however, fewer persons with HIV-1 are developing HAD and are instead experiencing either ANI or MND [161]. Neurocognitive impairments that precede HAD correlate with neuronal damage, and mechanisms by which HIV-1 induces synaptodendritic injury have been extensively reviewed [162]. Unsurprisingly, the molecular factors associated with neuronal loss, such as Ca^2+^ and mitochondrial instability, are also associated with changes in HIV-1 protein-induced changes in synaptic and dendritic morphology. HIV-1 protein Tat was found to increase cytosolic Ca^2+^ through NMDARs, which is largely responsible for Tat-induced dendrite swelling in mouse striatal neurons [163]. Additionally, gp120 induced cytoskeletal abnormalities, including rod-like complexes composed of cofilin and actin in hippocampal neurons, which was dependent on ROS formation by NADPH-oxidase 2 (NOX2) [164]. Key molecular factors thought to mediate neuronal loss, such as Ca^2+^ and ROS generation, are also shared in the pathways associated with synaptic injury, which are likely to underlie the neurocognitive impairment observed in the milder forms of HAND. Thus, targeting the upstream effects of these neurotoxic HIV-1 proteins may mitigate not only neuronal death but synaptodendritic injury as well.

## 6. Crosstalk between Inflammation and Mitochondrial Induced Cell Death

In this review, we have primarily covered two major pathways involved in the pathophysiology of neurodegeneration in NDs: NLRP3 inflammasome-mediated pyroptosis and Ca^2+^-mediated mitochondrial-induced apoptosis. These two pathways are not mutually exclusive, and multiple facets of crosstalk are present. For example, Ca^2+^ influx can directly or indirectly (via ER stress and ROS) activate NLRP3 inflammasomes. Additionally, inositol-requiring transmembrane kinase/endoribonuclease 1α (IRE1α), an ER stress sensor, regulates the release of inflammatory cytokines which can activate the NLRP3 inflammasome while also further producing ROS, which induces mitochondrial damage [165]. Crosstalk between ER stress and NLRP3 inflammasomes has been implicated in non-neuronal forms of cell death, such as cardiovascular disease [166] and renal ischemia [167]. In astroglia cells, ethanol-induced NLRP3 inflammasome activation was mediated by mitochondrial ROS generation. Interestingly, this study showed that ~73% of cell death was attributed to caspase-1-mediated pyroptosis, and the other ~25% was attributed to caspase-3-mediated apoptosis [168]. Both pyroptosis and apoptosis have been implicated as forms of cell death featured in the NDs discussed in this review. Therefore, therapeutic strategies for NDs should be developed to target upstream of the activation of these forms of cell death (such as disturbances in NT systems) to ensure that all forms of neuronal loss (and injury) can be prevented.

## 7. Conclusions

Different types of cell death, including (but not limited to) pyroptosis and apoptosis, are a featured hallmark in NDs. The machinery necessary for these forms of cell death is ubiquitously present in all cell types, which begs the question—why do some cells die and not others? More specifically, how is cell-specific loss/injury brought about in various NDs? We propose that the unique pattern of neurodegeneration evidenced in various NDs such as AD, PD, and HAD may be partially explained by their disruption of specific NT systems. In the context of NeuroHIV, extensive studies demonstrate a clear link between perturbation of monoaminergic transmission by exposure of the CNS to HIV-1 viral proteins and risk for development of HAND. Importantly, abused substances directly target NT transmission and exacerbate NT-mediated development of neurodegenerative diseases. Further work is needed to determine whether the viral protein-induced dysregulation of NT systems is sufficient to induce neuronal damage through the activation of downstream pathways, such as mitochondrial-mediated apoptosis or the release of inflammatory cytokines from macrophages to induce pyroptosis. Currently, there are no promising therapeutic strategies for neurodegenerative disease. Considering the progressive nature of neurodegenerative diseases, it is probably unsurprising that aging is a risk factor for NDs, and as populations live longer, NDs are becoming more prevalent [169]. It should be considered that the neuronal damage and/or loss induced by disturbances in NTs may be due to a smaller magnitude of mitochondrial stress (or inflammation) sustained over a long period of time rather than a large immediate effect as one might see for instance, with abused substances such as meth or cocaine [170]. Based on our current understanding of the neuropathology of NDs, establishing an early intervention strategy would be beneficial to prevent disturbances in NT transmission. For example, the effectiveness of an early therapeutic intervention for HAND to preserve neurocognitive functions in HIV-infected individuals may ultimately depend on a therapeutic strategy that combines compound(s) that specifically attenuate Tat binding site(s) in DAT with antiretroviral therapy without affecting the normal function of DAT [20,135]. In summary, this review sheds light on the impact of disturbances of NT systems on the development of various NDs.

## Figures and Tables

**Figure 1 ijms-24-15340-f001:**
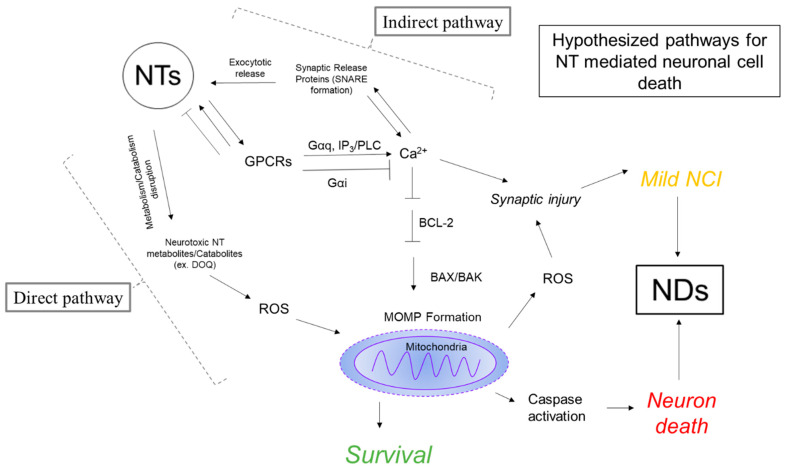
Two pathways are proposed for NT-mediated neuronal cell death and damage. The direct pathway involves neurotoxic metabolites or catabolites of NTs, which generate ROS, stimulating mitochondrial response. The indirect pathway involves the overactivation of G-protein coupled receptors to increase Ca^2+^ levels past physiologically normal levels, resulting in cellular stress responses and subsequent activation of mitochondrial-mediated apoptosis. Both pathways may facilitate synaptic injury through increased ROS formation. Molecular factors that contribute to the dysregulation of NT systems may mediate mild Neurocognitive Impairment (NCI) or neuronal cell death through the outlined pathways.

**Figure 2 ijms-24-15340-f002:**
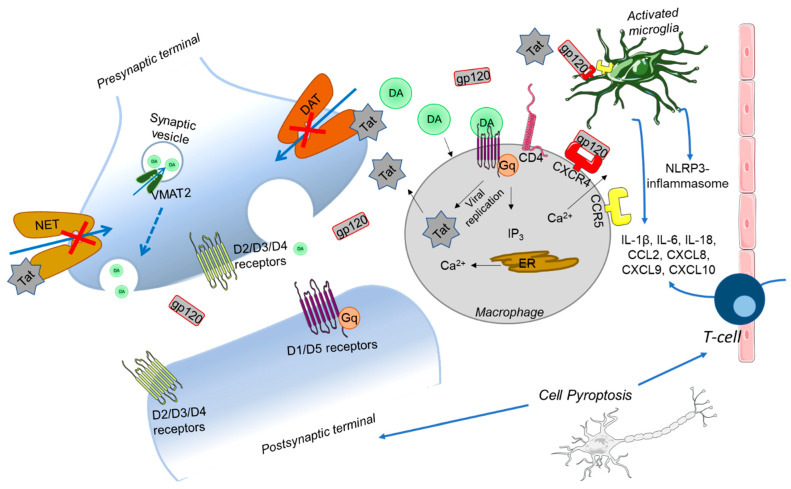
Hypothesized pathway for DA mediated HAND neurobiology. HIV-1 enters the CNS by infecting macrophages, which subsequently translocate across the blood–brain barrier. Once in the CNS, viral replication takes place in infected macrophages in the basal ganglia. HIV proteins such as Tat and gp120 are then released from infected cells into the extracellular space. Extracellular Tat can directly block DA and NE transporter activity (red “X”), subsequently increasing extracellular DA levels (dashed arrow). DA may then increase viral replication in macrophages and promote the release of inflammatory cytokines from macrophages, presenting a pathway by which the NT DA mediates neurotoxicity in subcortical brain regions containing dopaminergic neurons. Additionally, extracellular gp120 interacts with CXCR5 and CCR5 receptors in microglia and macrophages, which also contributes to neurotoxicity through activation of the NLRP3-inflammasome.

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
