# Peer review of "The Impact of Neurotransmitters on the Neurobiology of Neurodegenerative Diseases"

_ijms, 2023, doi:10.3390/ijms242015340_

Round 1

Reviewer 1 Report

The authors have chosen an interesting topic, however there are a few concerns that have to be addressed before considering for the publication.

1.   In the figure 1, authors have suggested hypothetical mechanisms for the neuronal death. The authors may elaborate the term Mild NCI for the first time use.

Secondly, authors have proposed the mechanism at the very beginning, but the proving of hypothetical proposal was not seen in the beginning. Rather authors started elaborating the regulation of the Neurotransmission.

Either it may be appropriate to move the Figure to the section where authors started proving or supporting the mechanism or may consider altering the appropriate sentences.

2.      Line 296, authors have already expanded the α-synuclein (αSyn) in the Line 65, which should be corrected. Similarly it applies to “β-Amyloid” line 47 and Line 298.

3.      Line 485 to line 495, entire sentences are in italics, should be unform throughout the manuscript.

4.      Line 346, “NTs” and Line 444, it’s “NT’s”, which seen  several places in the manuscript, should be corrected.  It also applies to “Ca2+” and “Ca2+” in the manuscript.

Needs extensive editing 

Author Response

  1.  In the figure 1, authors have suggested hypothetical mechanisms for the neuronal death. The authors may elaborate the term Mild NCI for the first-time use.

Secondly, authors have proposed the mechanism at the very beginning, but the proving of hypothetical proposal was not seen in the beginning. Rather authors started elaborating the regulation of neurotransmission.

Either it may be appropriate to move the Figure to the section where authors started proving or supporting the mechanism or may consider altering the appropriate sentences.

We thank the reviewer for their comments. We have added the definition for mild NCI to the figure 1 caption. Further, we have removed reference to figure 1 from the introduction chapter. Figure 1 is now first referenced in section 4.3.2.

  1. Line 296, authors have already expanded the α-synuclein (αSyn) in Line 65, which should be corrected. Similarly, it applies to “β-Amyloid” line 47 and Line 298.

The additional definitions of αSyn and β-Amyloid have been removed.

  1. Line 485 to line 495, entire sentences are in italics, should be unform throughout the manuscript.

We thank the reviewer for this comment, we have revised those sentences to normal font.

  1. Line 346, “NTs” and Line 444, it’s “NT’s”, which seen several places in the manuscript, should be corrected.  It also applies to “Ca2+” and “Ca2+” in the manuscript.

The manuscript has been updated such that NTs and Ca2+ are uniformly presented throughout.

Reviewer 2 Report

Overview of the manuscript

The work is a review focused on the impact of neurotransmitters, in particular dopamine and acetylcholine in neurodegenerative disease.

GENERAL  COMMENT

The work is interesting and focuses on a complex topic that can obtain from a review an adequate contribute in the understanding of the complex phenomenon of neurodegeneration. The focus on the HIV-Associated Neurocognitive Disorder contributes to enriching the work with an interesting and new topic. The iconographic details are clear and adequately illustrate the topics.

Architecturally the work is well performed in all its parts. Bibliographic citations are adequate in number and relevance. I suggest avoiding the oldest citations.

Specific comments

Pag. 11, line 485-492: change the italic in normal style.

I suggest the use of bibliographic citation non oldest than 1995-1994, or limit them to the case in which they are particular significative.

I suggest to substitute Levitt, 1965 and Christenson, 1972, with more actual citations.

Author Response

  1. Pag. 11, line 485-492: change the italic in normal style.

We thank the review for their helpful feedback. We have changed the italic style to normal.

  1. I suggest the use of bibliographic citation non oldest than 1995-1994 or limit them to the case in which they are particular significative.

We thank the reviewer for this comment, however, after thoroughly searching the document, we were unable to locate the sentences which are in italics which are being referenced.

We have removed references from before 1994 where appropriate.

  1. I suggest substituting Levitt, 1965 and Christenson, 1972, with more actual citations.

We thank the reviewers for this perspective and have replaced these references with a more updated reference – Meiser et al 2013.

Round 2

Reviewer 1 Report

Manuscript quality Improved

Minor editing of English language required